# Src Inhibition Attenuates Liver Fibrosis by Preventing Hepatic Stellate Cell Activation and Decreasing Connective Tissue Growth Factor

**DOI:** 10.3390/cells9030558

**Published:** 2020-02-27

**Authors:** Hye-Young Seo, So-Hee Lee, Ji-Ha Lee, Yu Na Kang, Jae Seok Hwang, Keun-Gyu Park, Mi Kyung Kim, Byoung Kuk Jang

**Affiliations:** 1Department of Internal Medicine, Keimyung University School of Medicine, Daegu 42601, Korea; seo568@hanmail.net (H.-Y.S.); jy16162727@naver.com (S.-H.L.); jihain10@gmail.com (J.-H.L.); gastro@dsmc.or.kr (J.S.H.); 2Institute for Medical Science, Keimyung University School of Medicine, Daegu 42601, Korea; 3Department of Pathology, Keimyung University School of Medicine, Daegu 42601, Korea; Yunakang@dsmc.or.kr; 4Department of Internal Medicine, School of Medicine, Kyungpook National University, Daegu 41944, Korea; kpark@knu.ac.kr

**Keywords:** Src, saracatinib, PP2, liver fibrosis, HSC, TGF-β, hepatocyte, CTGF, Smad3, autophagy

## Abstract

The SRC kinase family comprises non-receptor tyrosine kinases that are ubiquitously expressed in all cell types. Although Src is reportedly activated in pulmonary and renal fibrosis, little is known regarding its role in liver fibrosis. This study investigated whether the inhibition of Src protects against liver fibrosis. The expression of Src was upregulated in thioacetamide (TAA)-induced fibrotic mouse liver and cirrhosis of patients, and phospho-Src was upregulated during activation of hepatic stellate cells (HSC). In addition, Src inhibition reduced the expression of α-smooth muscle actin (αSMA) in primary HSCs and suppressed transforming growth factor β (TGF-β)-induced expression of connective tissue growth factor (CTGF) in hepatocytes. Src inhibitor Saracatinib also attenuated TAA-induced expression of type I collagen, αSMA, and CTGF in mouse liver tissues. The antifibrotic effect of Src inhibitors was associated with the downregulation of Smad3, but not of signal transducer and activator of transcription 3 (STAT3). In addition, Src inhibition increased autophagy flux and protected against liver fibrosis. These results suggest that Src plays an important role in liver fibrosis and that Src inhibitors could be treat liver fibrosis.

## 1. Introduction

Liver fibrosis is the end result of many chronic liver diseases with various etiologies [1]. The formation of connective tissue and deposition of extracellular matrix (ECM) proteins characterizes fibrosis [2,3]. Hepatic stellate cells (HSCs) play an important role in the initiation and progression of liver fibrosis. Upon liver injury, HSCs are activated and transdifferentiate into myofibroblasts, which produce excessive amounts of ECM proteins, such as α-smooth muscle actin (αSMA) [4,5]. Connective tissue growth factor (CTGF) is an ECM protein that is involved in many biological processes, such as proliferation, migration, and differentiation. CTGF expression was markedly increased in HSCs and transforming growth factor β (TGF-β)-stimulated hepatocytes, and CTGF was demonstrated as a major factor in the pathogenesis of tissue fibrosis [6,7]. In fibroblasts, TGF-β-induced CTGF is dependent on Smad3 and transducer and activator of transcription 3 (STAT3) are involved in activated CTGF [8]. The SRC kinase family comprises non-receptor tyrosine kinases, including Src, Fyn, Yes, and Lyn [9,10]. Src was originally identified as an oncogene and it is expressed in human tumors; however, it is also ubiquitously expressed in all cell types. Src phosphorylates the signaling proteins STAT3, AKT, and epidermal growth factor receptor (EGFR), thereby regulating various biological activities, including cell survival, proliferation, and migration [10,11,12]. The activation of Src causes pulmonary and renal fibrosis [13,14]. We recently demonstrated that the loss of Fyn, SRC family kinase, inhibits TGF-β-induced the expression of the αSMA in kidney cell lines and prevents unilateral ureteral obstruction (UUO) induced renal fibrosis [15]. These observations suggest that Src activation is a pathologic observation in fibrotic diseases. However, little is known rgarding the role of Src in liver fibrosis. This study investigated the role of Src in HSCs and liver fibrosis.

## 2. Materials and Methods

### 2.1. Materials

AstraZeneca UK Limited (London, UK) kindly provided the Src kinase inhibitor saracatinib (AZD0530). Recombinant human TGF-β (5 ng/mL) was purchased from R&D Systems (Minneapolis, MN, USA). Thioacetamide (TAA), PP2, and chloroquine were purchased from Sigma-Aldrich (St. Louis, MO, USA). PP2 was purchased from Millipore (Bedford, MA, USA). Anti-CTGF (SC-365970) and anti-phospho-Smad2/3 (SC11769) antibodies were purchased from Santa Cruz (Dallas, TX, USA). An anti-αSMA (A2547) antibody was purchased from Sigma-Aldrich. Anti-collagen (ab34710), anti-PAI-1 (ab66705), and anti-p62 (ab56416) antibodies were purchased from Abcam (Cambridge, UK). Anti-GAPDH (CS2118), anti-phospho-Src (Y416) (CS2101), anti-phospho-Smad3 (Ser423/425) (CS9520), anti-LC3B (CS2775), and anti-Atg7 (CS2631) antibodies were purchased from Cell Signaling Technology (Beverly, MA, USA).

### 2.2. Animal Study

Male 8-week-old C57BL/6 mice were purchased from Central Lab Animal (Seoul, Korea) and housed in a facility under a 12 h light/dark cycle. All experiments were approved by the Institutional Animal Care and Use Committee of Keimyung University (KM-2017-33). All animal procedures were performed in accordance with the institutional guidelines for animal research. Saracatinib was administered orally premixed with the milled pellet (FEEDLAB, Guri, Korea). TAA induces liver damage and fibrosis via various mechanisms. The mice were divided into four groups and fed chow (control, n = 6), fed chow supplemented with saracatinib (50 mg/kg, n = 4), fed chow and injected with TAA (250 mg/kg of body weight, n = 7), or fed chow that was supplemented with saracatinib and injected with TAA (n = 7). Liver fibrosis was induced by intraperitoneal injecting mice with TAA dissolved in saline thrice per week for 8 weeks. Body weight and food intake were measured thrice per week. After 8 weeks, the animals were euthanized and blood and liver tissue samples were collected.

### 2.3. Cell Culture

The AML12 mouse hepatocyte cell line was purchased from the American Type Culture Collection (Manassas, VA, USA). These cells were cultured in 5% CO_2_ and 95% air at 37 °C in DMEM/F12 (GIBCO-BRL, Grand Island, NY, USA) that was supplemented with insulin-transferrin-selenium (GIBCO-BRL), dexamethasone (40 ng/mL; Sigma-Aldrich), antibiotics (GIBCO-BRL), and 10% fetal bovine serum (FBS; Hyclone, Logan, UT, USA). LX2 human HSCs were a kind gift from Dr. Jeong (Korea Advanced Institute of Science and Technology, Daejeon, Korea). These cells were cultured in 5% CO_2_ and 95% air at 37 °C in DMEM (GIBCO-BRL) that was supplemented with antibiotics and 10% FBS. The cells were treated with chemicals in 0.5% FBS with or without TGF-β (5 ng/mL) and then processed for isolation of proteins, as described below.

### 2.4. Isolation of Primary Hepatocytes

The hepatocytes were isolated from C57BL/6 mice by the perfusion of the liver through the portal vein. The liver was perfused with resuspension buffer (5.4 mmol/L KCl, 0.44 mmol/L KH_2_PO_4_, 140 mmol/L NaCl, 0.34 mmol/L Na_2_HPO_4_, 0.5 mmol/L EGTA, and 25 mmol/L Tricine, pH 7.2) at a rate of 5 mL/min for 10 min, and then with collagenase solution (Ca^2+^ and Mg^2+^-free Hanks Balanced Salt Solution, pH 7.2, containing 0.75 mg/mL collagenase type I (Worthington Biochemical Corp., Freehold, NJ, USA)) at a rate of 5 mL/min for 10 min. After perfusion, the liver was shaken for 20 min at 37 °C and, filtered through a 70 µm nylon mesh and centrifuged at 42 × g for 5 min at 4 °C. Pelleted hepatocyte were resuspended in William’s medium E (GIBCO-BRL) and seeded onto type I collagen-coated 60 mm dishes (IWAKI Scitech Kiv, Tokyo, Japan). The viability of hepatocytes was always higher than 85%. After incubation for 2–3 h, the medium was replaced by medium 199 (Sigma-Aldrich). The hepatocytes were treated with chemicals in 0.5% FBS with or without TGF-β (5 ng/mL) and then processed for isolation of proteins, as described below.

### 2.5. Isolation of Primary HSCs

The HSCs were isolated from C57BL/6 mice by perfusion of the liver through the inferior vena cava. The liver was perfused with EGTA buffer (136.89 mmol/L NaCl, 5.37 mmol/L KCl, 0.64 mmol/L NaH_2_PO_4_·H_2_O, 0.85 mmol/L Na_2_HPO_4_, 9.99 mmol/L HEPES, 4.17 mmol/L NaHCO_3_, 0.5 mmol/L EGTA, and 5 mmol/L glucose, pH 7.35–7.4) at a rate of 5 mL/ for 2 min, followed by enzyme buffer (136.89 mmol/L NaCl, 5.37 mmol/L KCl, 0.64 mmol/L NaH_2_PO_4_·H_2_O, 0.85 mmol/L Na_2_HPO_4_, 9.99 mmol/L HEPES, 4.17 mmol/L NaHCO_3_, and 3.81 mmol/L CaCl_2_·2H_2_O, pH 7.35–7.4) containing 0.4 mg/mL pronase (Roche Diagnostics, Indianapolis, IN, USA) at a rate of 5 mL/min for 5 min and then with Enzyme buffer containing 0.193 U/mg collagenase (Roche Diagnostics) at a rate of 5 mL/min for 7 min. After perfusion, the liver was shaken for 25 min at 37 ℃, filtered through a 70 µm nylon mesh, and centrifuged at 580 × g for 10 min at 4 ℃. Pelleted HSCs were resuspended in Gey’s Balanced Salt Solution (GBSS) (Sigma-Aldrich), gently overlaid with a gradient of Cell-OptiPrep™ (Sigma-Aldrich) prepared with GBSS using a pipette, and then centrifuged at 1380 × g for 17 min at 4 °C without braking. HSCs present in a thin white layer at the interface between Cell-OptiPrep™ and GBSS were harvested and washed with Hank’s Balanced Salt Solution. The cells were plated in DMEM (GIBCO-BRL) containing 10% FBS. The medium was changed every 2 days.

### 2.6. Small Interfering RNA (siRNA)-Mediated Depletion of Src

Predesigned Src-targeting siRNA (siSrc) and control siRNA (siCon) were purchased from Santa Cruz. The cells were transfected with 100 nM siRNA while using Lipofectamine RNAiMAX (Invitrogen, Waltham, MA, USA) for 5 h, cultured in medium containing 0.5% FBS, and harvested ~48 h after transfection.

### 2.7. Western Blot Analysis

The cells were incubated in RIPA buffer (Thermo Scientific, Waltham, MA, USA) for 30 min at 4 °C. The mouse liver samples were homogenized and lysed in RIPA buffer. Protein concentrations were determined while using the BCA assay (Thermo Scientific). 10 μg of proteins were separated by SDS-PAGE and then transferred to membranes (Millipore, Burlington, MA,). Membranes were sequentially incubated in blocking buffer (5% skimmed milk prepared in Tris-buffered saline containing 0.1% Tween 20), primary antibodies (1:1000), and appropriate horseradish peroxidase-conjugated secondary antibodies. The signals were visualized while using a Clarity™ Western ECL substrate kit (Bio-Rad, Richmond, CA, USA). The membrane was re-probed with an anti-GAPDH antibody to verify that an equal amount of protein had been loaded in each lane. The signal intensities were quantitated by densitometry while using ImageJ software (NIH, Bethesda, MD, USA).

### 2.8. Quantitative Real-Time PCR

The total RNA was isolated from cells and tissue extracts while using TRIzol reagent (Invitrogen). Reverse transcription was performed using a Maxima First Strand cDNA Synthesis Kit (Thermo Scientific). Quantitative real-time RT-PCR was performed while using a SYBR Green PCR Master Mix Kit (Roche Diagnostics, Indianapolis, IN, USA)) and a CFX Connect Real-Time PCR system (Bio-Rad). The PCR conditions were, as follows: 45 cycles of 95 °C for 30 s, 60 °C for 10 s, and 72 °C for 15 s. The primer sequences are in Appendix A. GAPDH was used as an internal standard.

### 2.9. Immunohistochemical (IHC) Analysis

Liver tissue was fixed in 10% formalin, embedded in paraffin, and then cut into 4 µm thick sections. The liver sections were deparaffinized in xylene and rehydrated while using a graded ethanol series. For Sirius red staining, to evaluate collagen deposition, slides were immersed for 18 h in saturated picric acid with 0.1% Sirius red F3BA (Aldrich Chemicals). The slides were then washed in 0.5% acetin acid for 2 min and then dehydrated through graded alcohol concentrations. The slides were transferred to xylene, and the coverslip was mounted with Permount (Fisher Scientific, Edmonton, Alberta, Canada). IHC staining was performed while using anti-collagen, anti-αSMA, anti-CTGF, anti-PAI-1, and anti-phospho-Smad2/3 primary antibodies (1:500), followed by horseradish peroxidase-conjugated anti-mouse or anti-rabbit IgG secondary antibodies (Dako, Glostrup, Denmark), in accordance with the manufacturer’s instructions. All of the data were normalized against the equivalent data in mice fed chow (control). Immunostaining was quantified using ImageJ software (ImageJ software, 1.52a National Institutes of Health, Bethesda, MD, USA).

### 2.10. Atg7^flox/flox^ Albumin Cre Mice (Atg7^f/f^ Alb-Cre)

Atg7 floxed mice (Atg7^f/f^) were bred with albumin-Cre mice to generate Atg7 hepatocyte-specific knockout mice (Atg7^f/f^ Alb-Cre^+^). The mouse was a kind gift from Dr. Myung-Shik Lee (Yonsei University) with the permission of Dr. Masaaki Komatsu (Tokyo Metropolitan Institute of Medical Science). The Institutional Animal Care and Use Committee of Keimyung University approved all of the experiments (KM-2016-28). The genotyping of the offspring was performed by PCR, as described [16].

### 2.11. Immunofluorescence Analysis

AML12 cells were fixed in 10% formalin for 10 min and then permeabilized with 0.1% Triton-X100 for 15 min. The cells were imaged using a confocal microscope after incubation with anti-LC3 (1:500) and a Alexa Fluor conjugated secondary antibody (Abcam, Cambridge, UK), (Carl Zeiss, Oberkochen, Germany).

### 2.12. Patients and Specimens

Non-fibrotic and cirrhotic liver tissues were obtained from 11 patients with intrahepatic duct stones and 14 patients with cirrhosis who underwent hepatectomy at Keimyung University, Dongsan Medical Center, between January 2002 and December 2018. The Institutional Review Board of Keimyung University Dongsan Hospital approved this study (2019-01-009). Formalin-fixed and paraffin-embedded tissue blocks were selected for tissue microarray sampling. A puncher tip (5.0 mm diameter) was used to punch out representative non-fibrotic and cirrhotic areas. Three cores per case were used to construct a tissue microarray. IHC staining of Src was performed while using a rabbit anti-Src polyclonal antibody (Abcam, Cambridge, MA, USA) and an automatic staining device (Benchmark XT; Ventana Medical Systems, Mountain View, CA, USA), in accordance with the manufacturers’ protocols.

### 2.13. Statistical Analysis

The data are expressed as means ± SEM. Statistical analyses were performed while using a one-way ANOVA analysis with Duncan’s test or the two-tailed Student’s t test. *P* < 0.05 was considered to be statistically significant. All of the experiments were performed at least three times.

## 3. Results

### 3.1. Src is Upregulated in Liver Tissues of TAA-Injected Mice and Cirrhotic Livers of Patients

First, we examined activation of SRC family kinases in the mouse model of TAA-induced liver fibrosis. Src mRNA expression was significantly upregulated in the liver tissues of TAA-injected mice; however, mRNA expression of other Src family kinases was not significantly altered (Figure 1A). Moreover, the levels of phospho-Src (Y416) and total-Src were significantly increased in the liver tissues of TAA-injected mice (Figure 1B). IHC staining confirmed that the level of total Src was significantly increased in liver tissues of these mice (Figure 1C). Next, we investigated whether Src is upregulated in pathologically fibrotic human livers. IHC staining of total Src revealed that Src expression was significantly higher in the liver tissues of patients with liver cirrhosis than in liver tissues of normal controls (Figure 1D). These results indicate that Src plays an important role in the fibrotic liver.

### 3.2. Src is Involved in Hepatic Stellate Cell Activation and TGF-β Stimulation

We examined Src expression during the activation of HSCs because HSCs activation is involved in the progression of liver fibrosis. To this end, we activated freshly isolated quiescent HSCs by culturing them for 7 days. The expression of αSMA and phospho-Src increased during the activation of primary HSCs (Figure 2A). We performed siRNA targeting Src to determine whether Src mediates HSCs activation. The suppression of Src inhibited αSMA expression on the 7 day of HSCs culture, as shown in Figure 2B. Next, we investigated whether Src is activated in cells treated with TGF-β. TGF-β treatment (5 ng/mL) induced the phosphorylation of Src in LX2 cells at up to 8 h and in primary hepatocytes and AML12 cells at 1–2 h (Figure 2C–E). Moreover, TGF-β treatment increased PAI-1 expression in LX2 cells and CTGF expression in hepatocytes (Appendix A). We depleted endogenous Src using siSrc to determine whether Src mediates TGF-β-induced CTGF expression. The depletion of Src significantly attenuated TGF-β-induced CTGF expression in primary hepatocytes (Figure 2F). These results show that the phosphorylation of Src plays an important role in the activation of HSC and it is associated with the expression of CTGF in hepatocyte.

### 3.3. Src Inhibition Attenuates TAA-Induced Liver Fibrosis and Activation of HSCs

Next, we investigated whether Src inhibitor saracatinib protects from TAA-induced liver fibrosis. It was confirmed that saracatinib attenuated the phosphorylation of Src in liver tissues (Appendix A). Sirius red staining revealed markedly elevated fibrosis following TAA treatment. In contrast, TAA-induced liver fibrosis was significantly decreased by saracatinib treatment. Furthermore, immunohistochemical (IHC) staining showed that the saracatinib-treated group had significantly reduced hepatic expression levels of type I collagen, PAI-1, and α-SMA (Figure 3). Real-time PCR and western blot analysis also revealed that the expression of collagen, PAI-1, and α-SMA were increased in the TAA-treated group, and saracatinib treatment prevented this effect (Figure 4A,B). We cultured primary HSCs and treated saracatinib while HSC was activated because the reduction of α-SMA contributes to the reduction of HSCs activation. Saracatinib inhibited αSMA expression in HSCs, thus confirming that activation of HSCs was inhibited, as shown in Figure 4C. Taken together, these data demonstrate that Src inhibition protects against TAA-induced liver fibrosis and the activation of HSCs.

### 3.4. Src inhibition Attenuates CTGF Expression

CTGF is expressed in fibrotic livers and it is a marker of liver fibrosis. Next, we investigated whether the Src inhibitors saracatinib and PP2 inhibit CTGF expression in the liver. IHC staining demonstrated that CTGF was significantly upregulated in the liver tissues of TAA-treated mice, and this effect was significantly attenuated by saracatinib (Figure 5A). Consistently, saracatinib and PP2 markedly attenuated TGF-β-induced CTGF expression in AML12 cells and primary hepatocytes (Figure 5B,C), as well as TGF-β-induced PAI-1 and fibronectin expression in LX2 cells (Figure 5D).

### 3.5. Src Inhibition Attenuates Phosphorylation of Smad3, but not of STAT3

The activation of the STAT3 and Smad3 pathways is implicated in CTGF expression [8,17]. Src inhibitors reportedly affect the STAT3 and Smad3 signaling pathways. Next, we investigated whether Smad3 mediates the inhibitory effects of saracatinib on liver fibrosis. The phosphorylation of Smad3 and its subsequent nuclear translocation are critical steps in the signaling cascade underlying liver fibrosis. We investigated the effect of saracatinib on the nuclear translocation of phospho-Smad3. IHC staining showed that phospho-Smad2/3 accumulated in nuclei in liver tissues of TAA-injected mice and this effect was significantly attenuated by saracatinib (Figure 6A). Consistently, saracatinib and PP2 attenuated TGF-β-induced phospho-Smad3 expression in AML12 cells, primary hepatocyte, and LX2 cells (Figure 6B–D). We previously reported that the inhibition of Fyn decreases STAT3 phosphorylation [15]; therefore, we investigated whether saracatinib inhibits STAT3 phosphorylation. Saracatinib did not attenuate TAA-induced phospho-STAT3 expression in the liver tissues of mice (Appendix A). Saracatinib attenuated TGF-β-induced phospho-STAT3 expression in AML12 cells and primary hepatocytes, but not in LX2 cells (Appendix A). PP2 did not attenuate TGF-β-induced phospho-STAT3 expression (Appendix A). These results suggest that saracatinib mainly inhibits liver fibrosis by attenuating the phosphorylation of Smad3.

### 3.6. Src Inhibition Prevents Liver Fibrosis through Autophagy Induction

Previously, SRC inhibitors have been reported to induce autophagy in cancer cells [18] and an increase in autophagy has an inhibitory effect on liver fibrosis [19]. Next, we examined the role of autophagy in the prevention of liver fibrosis by saracatinib and PP2. Saracatinib and PP2 induced the formation of LC3 puncta (Figure 7A), and chloroquine markedly increased LC3 II by saracatinib and PP2 (Figure 7B). These results suggest that saracatinib and PP2 increase autophagy flux. We used hepatocyte-specific Atg7-knockout mice to investigate the association between autophagy and liver fibrosis. In primary hepatocytes of Atg7^f/f^-Cre^+^ mice, Atg7 expression and conversion of LC3 I to LC3 II were not observed, and p62 protein expression was increased, thus confirming that autophagy did not occur (Figure 7C). Saracatinib did not decrease CTGF expression in primary hepatocytes of Atg7^f/f^-Cre^+^ mice (Figure 7D). This result suggests that saracatinib decreases the expression of CTGF by inducing autophagy.

## 4. Discussion

This study shows that Src is involved in the activation of HSCs and liver fibrosis. Src inhibition reduced the expression of αSMA and CTGF in primary HSCs and suppressed TGF-β-induced expression of CTGF in hepatocytes. Saracatinib also attenuated TAA-induced liver fibrosis. The protective effect of Src inhibition against liver fibrosis was associated with the attenuation of TGF-β-induced phospho-Smad3 expression, but not with the attenuation of STAT3 phosphorylation. Moreover, the inhibition of CTGF expression by saracatinib in hepatocytes involved the induction of autophagy.

The activation of Src has been implicated in the pathogenesis of tumors and fibrosis in the lungs and kidneys [13,20]. However, few studies have investigated the role of Src in liver fibrosis. A recent report showed that phosphorylation of Src at Y416 and Y530 is decreased and increased during liver fibrosis, respectively [12]. However, the current study demonstrated that phosphorylation of Src at Y416 was increased during the activation of primary HSCs and in liver tissues of TAA-injected mice. Moreover, Src expression was increased in the livers of patients with liver cirrhosis. Li et al. reported that the SRC family kinase Lyn is highly expressed in a liver fibrosis model and promotes the activation of HSCs [21], thus suggesting that increased expression of SRC family kinases is associated with the activation of HSCs and liver fibrosis.

TGF-β is a major mediator of ECM protein accumulation during liver fibrosis. In this study, TGF-β treatment increased the expression of phospho-Src in LX2 cells and hepatocytes. To confirm that Src plays an important role in liver fibrosis, we measured CTGF expression in primary hepatocytes that were transfected with siSrc. The knockdown of Src attenuated TGF-β-induced CTGF expression, suggesting that Src activity is related to CTGF expression.

The Src inhibitor saracatinib has been tested in clinical trials for various types of cancer [22] and it was recently reported to inhibit lung fibrosis as well as cancer [13,23]. PP1 and PP2, which inhibit Src family kinases, also attenuate the expression of ECM proteins in cultured renal cells [15,20]. In this study, saracatinib and PP2 were used as Src inhibitors. Saracatinib attenuated TAA-induced liver fibrosis in mice and inhibited αSMA expression during the activation of HSCs, while saracatinib and PP2 attenuated TGF-β-induced CTGF expression.

Although the mechanism underlying Src-mediated liver fibrosis is not fully understood, the signaling pathways that are responsible for Src activation and the targets of activated Src have been extensively studied. It has been reported that Src induces the phosphorylation of STAT3 and Smad3 activated by TGF-β [24]. Similar to the results of other study [15], saracatinib did not decrease the TGF-β mRNA level in this study (Appendix A). However, saracatinib inhibited TGF-β-induced phospho-Smad3 expression in AML12 cells, primary hepatocytes, and LX2 cells. Saracatinib also inhibited TAA-induced Smad3 translocation to the nucleus. On the other hand, the increase in phospho-STAT3 in liver tissues of TAA-injected mice was not reduced by saracatinib, and PP2 did not decrease the level of phospho-STAT3 in vitro (Appendix A). We reported that Fyn deficiency inhibits renal fibrosis by reducing the expression of phospho-STAT3, but not of Smad3 [15]. These results suggest that the antifibrotic effect of saracatinib that is induced by the inhibition of CTGF in the liver is due to the inhibition of Smad3, unlike the mechanism in the kidney. The ERK and JNK signaling pathways are also involved in expression of CTGF induced by TGF-β [25,26]. In this study, saracatinib attenuated phospho-JNK and phospho-ERK expression in the liver tissues of TAA-injected mice (Appendix A).

According to recent reports, Src-mediated EGFR signaling is required for the induction of angiotensin-II by TGF-β, and sustained EGFR activation leads to the production of growth-promoting growth factors, such as TGF-β [27,28]. In the current study, Src inhibition attenuated TGF-β-induced phospho-EGFR expression in AML12 cells and primary hepatocytes (Appendix A); however, the effect of this inhibition on TAA-induced liver fibrosis is unclear. Further research is required to clarify this point. Autophagy is an intracellular pathway of catabolism that targets defective cell bodies and excess components to lysosomes for degradation [29]. Recent studies have demonstrated that autophagy is associated with liver fibrosis [19,30] and autophagy-enhancing drug reduces the hepatic load and improves liver fibrosis [31]. Saracatinib and PP2 also induce autophagy, which protects against apoptosis [18]. In this study, saracatinib and PP2 induced autophagy flux. However, saracatinib did not decrease CTGF expression in primary hepatocytes of Atg7^f/f^-Cre^+^ mice, in which autophagy was completely blocked. These results suggest that autophagy regulates CTGF expression in hepatocytes. The effect of such autophagy was unclear in HSCs and animal experiments, so it is difficult to conclude what role it plays in liver fibrosis. Therefore, further experiments on the role of autophagy are needed in the future.

In conclusion, our results show that Src is upregulated during the activation of HSCs and liver fibrosis. The Src inhibition prevented TAA-induced liver fibrosis, inhibited the activation of HSCs, and significantly reduced TGF-β-induced CTGF expression. Thus, Src is a potential therapeutic target in liver fibrosis.

## Figures and Tables

**Figure 1 cells-09-00558-f001:**
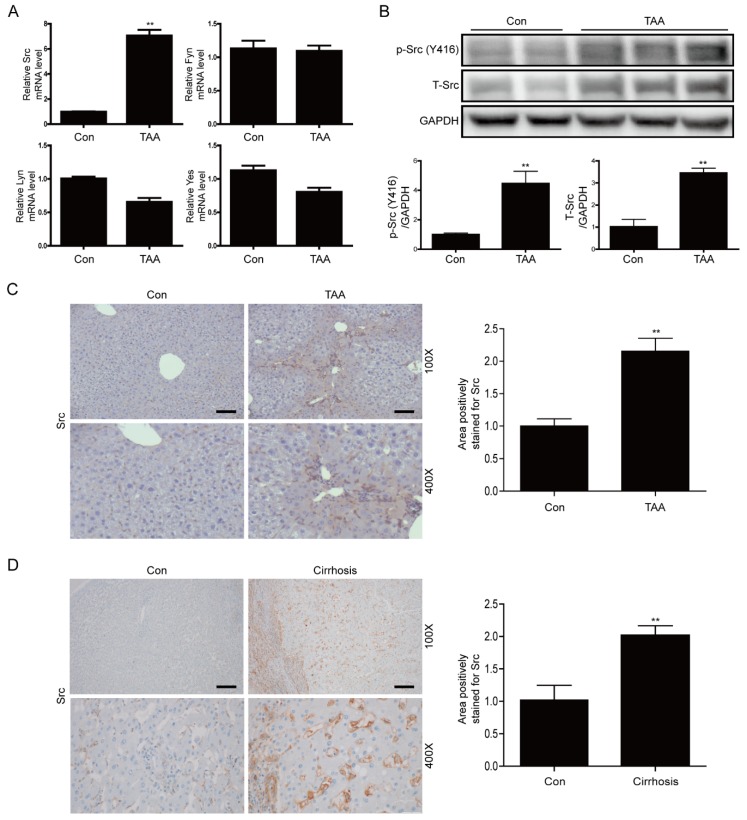
Expression of Src is elevated in liver tissues of thioacetamide (TAA)-injected mice and cirrhotic livers of patients (**A**) Representative real-time RT-PCR analysis of mRNA expression of SRC family kinases (Src, Fyn, Lyn, and Yes) in liver tissues of TAA-injected mice. Data in the bar graphs are means ± SEM. ** *p* < 0.01 compared with the control (Con). (**B**) Representative western blot analysis of Src and phospho-Src in liver tissues of TAA-injected mice. Data in the bar graphs are means ± SEM. ** *p* < 0.01 compared with control (Con). (**C**) Representative images of IHC staining for Src in liver tissues of TAA-injected mice. Areas of positive Src immunostaining were quantified by ImageJ software. All morphometric data of TAA-injected mice livers were normalized against those of the control, and the data in all bar graphs are expressed as fold increases relative to the control. Data in the bar graph are means ± SEM. ** *p* < 0.01 compared with control (Con). Original magnification ×100, ×400. Scale bars indicate 100 μm. (**D**) Representative images of IHC staining for Src in cirrhotic liver. Areas of positive Src immunostaining were quantitated by ImageJ software. All morphometric data obtained in cirrhotic liver were normalized against the corresponding values in control (Con), and the data in all bar graphs are expressed as the fold increase relative to the control. Data in the bar graph are means ± SEM. ** *p* < 0.01 compared with the control. Original magnification ×100, ×400. Scale bars indicate 100 μm.

**Figure 2 cells-09-00558-f002:**
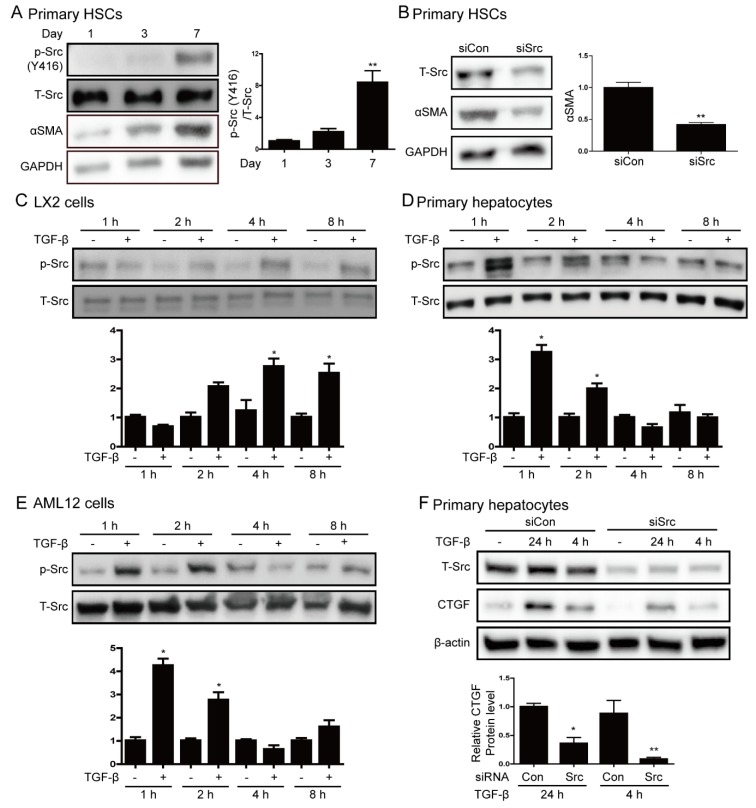
Src phosphorylation is increased by hepatic stellate cell activation and transforming growth factor β (TGF-β) stimulation. (**A**) Representative western blot analysis of phospho-Src and αSMA during activation of HSCs. Primary HSCs were cultured in DMEM containing 10% FBS. Data in the bar graphs are means ± SEM. ** *p* < 0.01 compared with one day. (**B**) Western blot analysis of the effect of Src depletion on αSMA expression. HSCs cultured for one day were transfected with 100 nM siSrc or control siRNA (siCon) and harvested at sven days. Data in the bar graphs are means ± SEM. ** *p* < 0.01 as compared with siCon. (**C**–**E**) LX2 cells, primary hepatocytes, and AML12 cells were treated with 5 ng/mL TGF-β for the indicated durations. Expression of phospho-Src (Y416) was investigated by western blot analysis. Data in the graph are represented as the mean ± SEM of three independent measurements. * *p* < 0.05 compared with control for each time. (**F**) Western blot analysis of the effect of Src depletion on TGF-β-induced connective tissue growth factor (CTGF) protein expression. Primary hepatocytes were transfected with 100 nM siSrc or control siRNA (siCon), and then treated with or without TGF-β. Data in the bar graph are means ± SEM of three independent measurements. * *p* < 0.05, ** *p* < 0.01 as compared with TGF-β treatment plus control siRNA.

**Figure 3 cells-09-00558-f003:**
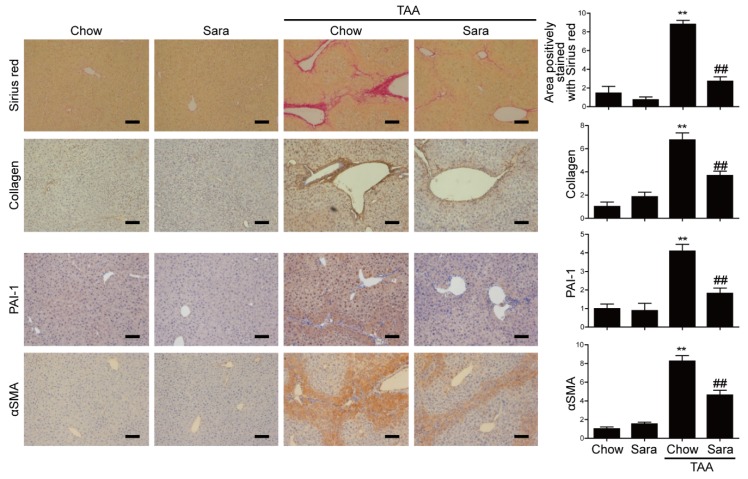
Saracatinib prevents TAA-induced liver fibrosis. Representative images of Sirius red staining and IHC staining with antibodies against type I collagen, PAI-1 and αSMA in liver tissue sections of TAA-injected mice treated with or without saracatinib. Data are means ± SEM of five random fields for each liver. ** *p* < 0.01 compared with chow group, ^##^
*p* < 0.01 compared with the TAA-injected chow group. Original magnification × 100. Scale bars indicate 100 μm.

**Figure 4 cells-09-00558-f004:**
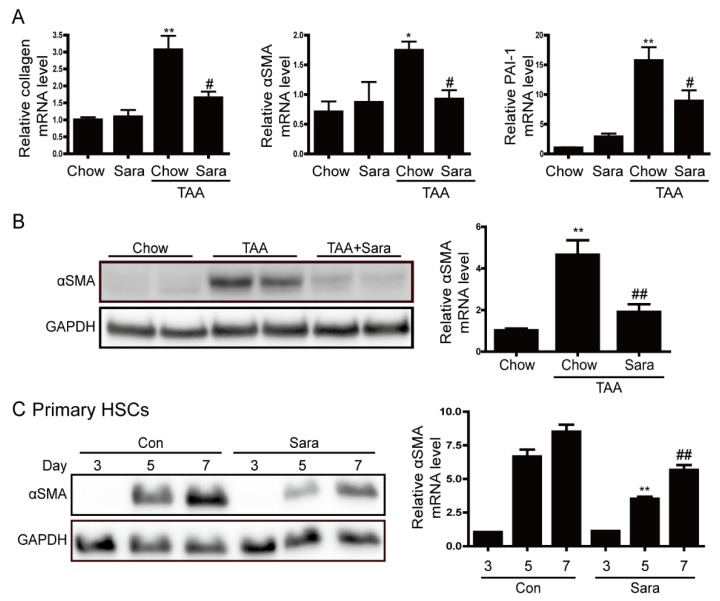
Saracatinib prevents TAA-induced liver fibrosis and inhibits αSMA expression in primary hepatic stellate cells (HSCs). (**A**) Representative real-time RT-PCR analysis of type I collagen, αSMA, and PAI-1 mRNA expression in liver tissues of TAA-injected mice treated with or without saracatinib. * *p* < 0.05, ** *p* < 0.01 compared with chow group, ^#^
*p* < 0.05 as compared with the TAA-injected chow group. (**B**) Representative western blot analysis of αSMA expression in liver tissues of TAA-injected mice treated with or without saracatinib. Data in the bar graph are means ± SEM. ** *p* < 0.01 compared with the chow group. ^##^
*p* < 0.01 compared with the TAA-injected chow group. (**C**) Western blot analysis of αSMA expression in cultured HSCs. Primary HSCs were cultured for 2 h, after which unattached cells and debris were removed by washing. HSCs were further cultured for three, five, and 7 days in DMEM containing 0.5% FBS with or without saracatinib. Data in the graph are represented as the mean ± SEM of three independent measurements. ** *p* < 0.01 compared with Day 5, ^##^
*p* < 0.01 compared with Day 7.

**Figure 5 cells-09-00558-f005:**
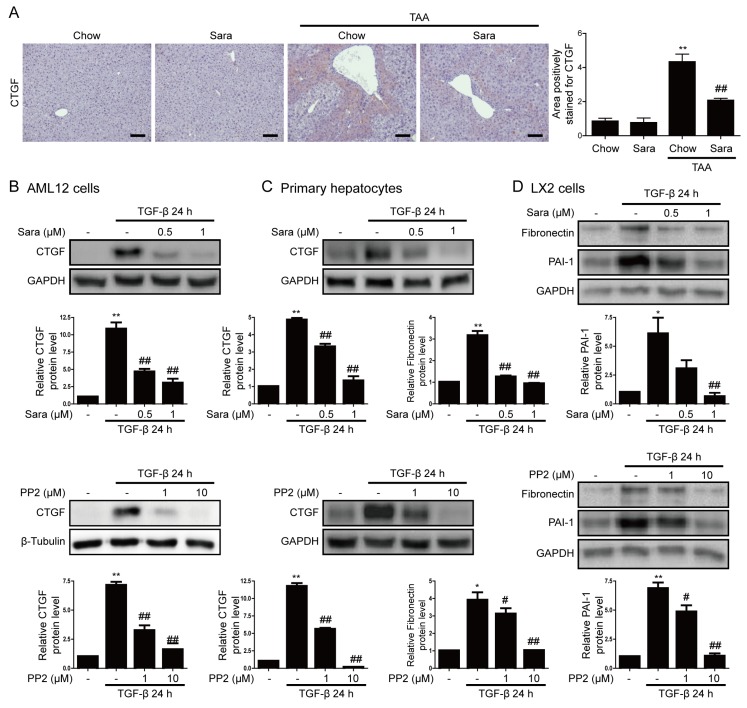
Src inhibitors attenuate CTGF expression. (**A**) Representative images of IHC staining for CTGF in livers of TAA-injected mice treated with or without saracatinib. Original magnification, ×100. All data were normalized against the corresponding values in control animals. Data in the bar graph are expressed as fold increase relative to the control. Data are means ± SEM of five random fields for each liver. ** *p* < 0.01 compared with the chow group, ^##^
*p* < 0.01 as compared with the TAA-injected chow group. (**B**–**D**) Western blot analysis of the effects of saracatinib and PP2 on TGF-β-induced CTGF and PAI-1 expression in AML12 cells, primary hepatocytes, and LX2 cells. Data in the graph are represented as the mean ± SEM of three independent measurements. * *p* < 0.01, ** *p* < 0.01 compared with the control, ^#^
*p* < 0.05, ^##^
*p* < 0.05 compared with TGF-β.

**Figure 6 cells-09-00558-f006:**
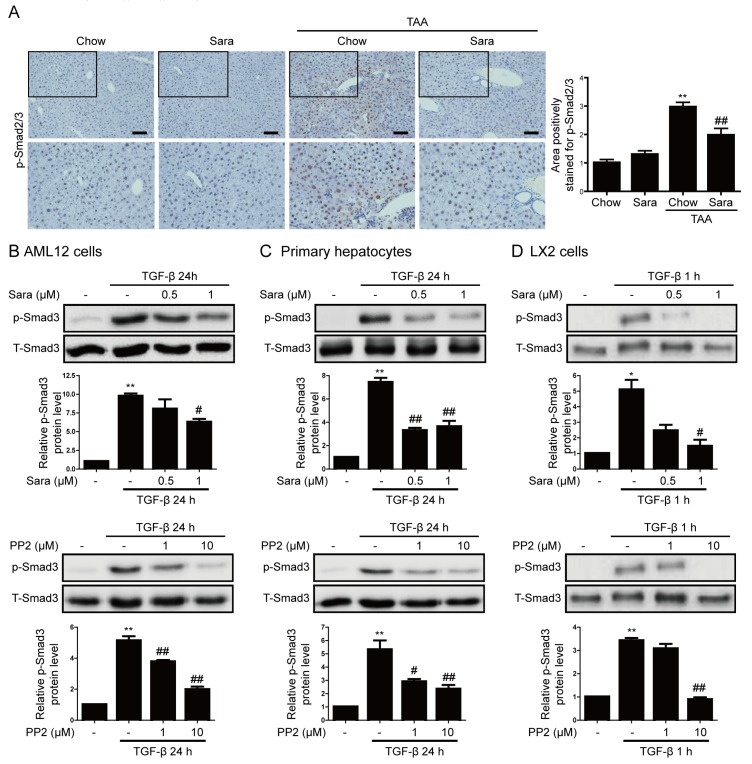
Src inhibitors attenuate Smad3 phosphorylation in liver tissues of TAA-injected mice and TGF-β-treated cells. (**A**) Representative images of IHC staining for phospho-Smad2/3 in the livers of TAA-injected mice treated with or without saracatinib. Original magnification, ×100. All data were normalized against the corresponding values in control animals. Data in the bar graph are expressed as fold increases relative to the control. Data are means ± SEM of five random fields for each liver. ** *p* < 0.01 compared with the chow group, ^##^
*p* < 0.01 compared with the TAA-injected chow group. (**B**–**D**) Western blot analysis of the effect of saracatinib and PP2 on TGF-β-induced phospho-Smad3 expression in AML12 cells, primary hepatocytes, and LX2 cells. Data in the graph are represented as the mean ± SEM of three independent measurements. * *p* < 0.01, ** *p* < 0.01 compared with the control, ^#^
*p* < 0.05, ^##^
*p* < 0.05 compared with TGF-β.

**Figure 7 cells-09-00558-f007:**
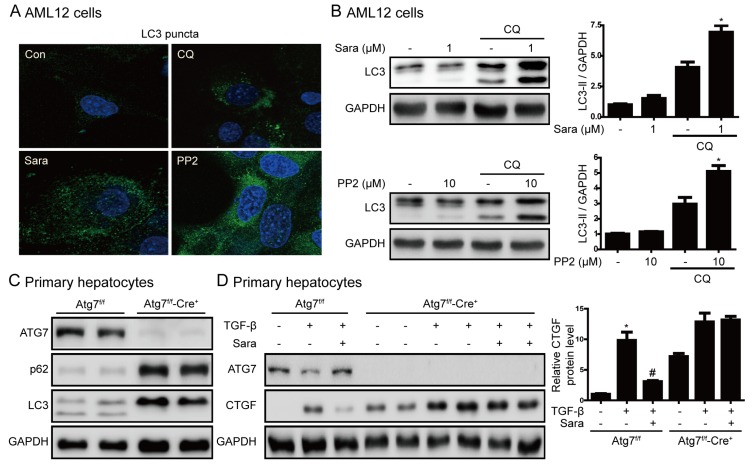
Induction of autophagy is required for prevention of liver fibrosis by saracatinib. (**A**) AML12 cells were treated with saracatinib, PP2, or chloroquine for 24 h and LC3 puncta formation was analyzed by immunofluorescence. Original magnification ×800. (**B**) Representative western blot showing the effects of saracatinib (1 μM), PP2 (10 μM), and chloroquine (CQ, 10 μM) on LC3 protein levels in AML12 cells. The data in the graph are represented as the mean ± SEM of three independent measurements. * *p* < 0.05 compared with CQ. (**C**) Western blot analysis of primary hepatocytes isolated from Atg7^f/f^-Cre^+^ mice showing decreased expression of ATG7 and LC3II and increased p62 levels. (**D**) Western blot analysis of the effect of saracatinib on TGF-β-induced CTGF expression in primary hepatocyte of Atg7^f/f^-Cre^+^ mice. Data in the graph are represented as the mean ± SEM of three independent measurements. * *p* < 0.05 compared with the control, ^#^
*p* < 0.05 compared with TGF-β.

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
