# Peer review of "Src Inhibition Attenuates Liver Fibrosis by Preventing Hepatic Stellate Cell Activation and Decreasing Connective Tissue Growth Factor"

_cells, 2020, doi:10.3390/cells9030558_

Round 1

Reviewer 1 Report

The manuscript presents novel work related to the involvement of Src, SMAD3, and autophagy in hepatic fibrosis. The paper features a comprehensive range of cell and animal models to support this relationship. The manuscript is comprehensive, interesting and appears fundamentally sound.

Minor Revisions/Suggestions:

Did you examine autophagy in primary hepatic stellate cells/LX2 cells? Autophagy seems to be associated with the fibrotic activation of hepatic stellate cells (see: Autophagy. 2012 Jan;8(1):126-8), therefore it would be interesting to know what saracatinib does to stellate cells. Activating autophagy in stellate cells could be counterproductive. Have you considered using the Smad3 Inhibitor, SIS3 and then looking at CTGF/autophagy markers? This would provide solid support for the direct involvement of SMAD3 in the signalling pathway. In some experiments it is unclear how many replicates have been performed for each experiment. Ideally n=x should be added to the results and/or the figure legends. In the abstract, it would be useful to state that TAA incudes fibrosis and that saracatinib is a Src inhibitor. This will help people outside the field quickly understand the abstract. If needed, the real-time PCR Primers could be placed in a supplementary table. Figure 1C should have 100X and 400X labels like Figure 1D for consistency. As the paper assembles a framework of a pathway, it might be useful to have a summary pathway figure in the discussion. Obviously this would have some parts missing, however this would help

Reviewer 2 Report

This manuscript by Seo et al. renders insightful notion to Src plays an important role in liver fibrosis and that Src inhibitors are therapeutically useful for liver fibrosis via delicate experimental settings. Nevertheless, a couple of issues should be detailed. Some descriptions regarding results require revision.

Abstract: The results of Src expression in patient cirrhotic liver (Fig. 1) represents clinical significance, and should be described in Abstract.

Fig. 2: Please specify the rationale behind the measurement of PAI-I.

Fig. 2 CDE: Please present densitometric data and annotate its statistic significance.

Fig 3. The Src expression in the liver of Chow, Sara, Chow+TAA, and Sara+TAA should be examined.

Fig. 3C: Please provide densitometric data and statistic annotation of alpha-SMA blotting data.

Fig. 4BCD: Please provide densitometric data and statistic annotation of the WB data.

Fig. 5B: Please provide densitometric data and statistic annotation of the WB data.

Fig. 6BCD: Can author simultaneously show Atg7, p62, and CTGF?

Fig. 6: Can author employ in vivo experimental setting with Atg7f/f-Cre+ miceto verify the role of Atg7 in mediating anti-fibrotic effect of saracatinib?

Reviewer 3 Report

As suggested by the editor, in order to receive a fast response I send the PDF with the comments and suggestions for the paper. 

The paper is well written, and the information provided in the paper could be relevant for exploring the development of new treatments in liver fibrosis. 

Seo et al describes the involvement of the Src in the development of liver fibrosis. They employ several primary and hepatic cells lines in the work, as well as animal and human samples. They also characterize some molecular mechanisms  implicated. 

All the comments for the authors are included in the PDF file.

Round 2

Reviewer 2 Report

The queries have been addressed by authors. It is a nice work to be been accepted.